# GRAPH2SEQ: SCALABLE LEARNING DYNAMICS FOR GRAPHS

## ABSTRACT

Neural networks are increasingly used as a general purpose approach to learning algorithms over graph structured data. However, techniques for representing graphs as real-valued vectors are still in their infancy. Recent works have proposed several approaches (e.g., graph convolutional networks), but as we show in this paper, these methods have difficulty generalizing to large graphs. In this paper we propose GRAPH2SEQ, an embedding framework that represents graphs as an infinite time-series. By not limiting the representation to a fixed dimension, GRAPH2SEQ naturally scales to graphs of arbitrary size. Moreover, through analysis of a formal computational model we show that an unbounded sequence is necessary for scalability. GRAPH2SEQ is also reversible, allowing full recovery of the graph structure from the sequence. Experimental evaluations of GRAPH2SEQ on a variety of combinatorial optimization problems show strong generalization and strict improvement over state of the art.

## 1 INTRODUCTION

Graphs are widely used to model pairwise relationships between objects in many real-world problems (e.g., gene interactions (Özgür et al., 2008), social networks such as Facebook (Ugander et al., 2011), etc.). Designing algorithms over graphs is therefore a topic of first order interest in many fields and applications. Today, most graph algorithms are designed by human experts. Unfortunately, in many problems, designing graph algorithms with strong performance guarantees is very challenging. Problems often involve messy optimization tasks with many constraints, or the current algorithmic understanding is simply limited (e.g., approximation gaps in CS theory literature).

In recent years, deep learning has emerged as a general-purpose toolbox for a variety of challenging learning tasks, from object recognition to language translation to learning complex heuristics directly from data. It is thus natural to ask whether we can apply deep learning to challenging graph-based optimization problems. The main challenge, however, is that the graph-structured data first needs to be *embedded* in a high dimensional Euclidean space before it can be used in a deep learning framework. *Graph representation* refers to the problem of representing graphs in Euclidean spaces.

Recent works have proposed many representation techniques for graphs (Bruna et al., 2013; Monti et al., 2016). Notable are a family of representations called graph convolutional neural networks (GCNN), with *spatial* and *spectral* variants, that attempt a representation by drawing inspiration from convolutional neural networks for images (Dai et al., 2017; Niepert et al., 2016; Defferrard et al., 2016). Some GCNN representations are for signals on a specific graph and some for varying sized graphs, but a deeper understanding of the similarities and differences between these representations and what they capture is largely an open question, with no consensus on one accepted technique (empirically or mathematically). Further, a key challenge is designing a representation that can scale to graphs of arbitrary shapes and sizes (Bruna & Li, 2017; Bronstein et al., 2017).

In this paper we propose GRAPH2SEQ, a novel method that represents graphs as a *time-series*. We show that our representation is a *strict generalization* over GCNNs, which typically represent graphs using fixed-dimensional vectors. Further, we show that GCNN variants and GRAPH2SEQ are all examples of a new computation model over graphs that we call LOCAL-GATHER, providing for a conceptual and algorithmic unification. Finally, we show that GRAPH2SEQ is information-theoretically lossless, i.e., the graph can be fully recovered from its time-series representation. We do this by making mathematical connections between GRAPH2SEQ and cause-effect relationship

studies among interacting entities by observing the dynamics of generated data (Granger, 1980; Rahimzamani & Kannan, 2016; Quinn et al., 2015).

The main advantage of GRAPH2SEQ is that one can harness the recent successes of recurrent neural network (RNN) architectures that make inferences on time-series data. We illustrate the power of this approach by combining GRAPH2SEQ with an appropriate RNN architecture and using it to tackle three classical combinatorial optimization problems, *minimum vertex cover, max cut* and *maximum independent set*, using reinforcement learning. On these combinatorial problems of varying complexity-theoretic hardness, we show that GRAPH2SEQ performs very well compared to state-of-the-art heuristics and shows significantly better performance and generalization than previous GCNN representations in the literature.

**Summary of Results**.
*(1)* GRAPH2SEQ *framework.* We propose a general *invertible* technique for representing arbitrary graphs as an infinite time-series. GRAPH2SEQ naturally combines with RNN architectures and reinforcement learning frameworks for downstream graph-optimization applications resulting in new state-of-the-art empirical performance.
*(2) Fundamental limits.* We define a formal computational model, that encapsulates GRAPH2SEQ as well as various GCNN representations in the literature, and study its fundamental limits. We show that fixed-length representations are fundamentally limited in their capabilities and are strictly inferior to the infinite time-series representation.
*(3) Evaluations.* We demonstrate the representation capability of GRAPH2SEQ in conjunction with a learning architecture involving RNNs by computing the minimum vertex cover, max cut and maximum independent sets of a graph. These are well-known NP-hard problems with varying hardness of approximations. We train on a *single, adversarially chosen* family of graphs of size 15, and demonstrate generalization to graphs of much larger sizes (example: 800) and across diverse graph structures. The adversarial choice of the training set and the *dynamically varying testing architecture* are the key deep learning innovations of this paper and are perhaps of broader interest.
*(4) Semantics of representation and learning.* We provide a coherent semantic understanding of GRAPH2SEQ's dynamics, both during test and training stages and present techniques that help visualize the dynamics in the learned model.

## 2 RELATED WORK

Our work is at the intersection of two topical areas of deep learning.

**Neural networks on graphs.** Early works to apply neural network based learning on arbitrary graphs are Gori et al. (2005); Scarselli et al. (2009). They consider an information diffusion mechanism, in which nodes update their states until an equilibrium is reached. Li et al. (2015) propose a variant that use gated recurrent units to perform the state updates, and has some similarity to our representation dynamics; however the sequence length does not vary between training and testing nor do the authors identify the sequence itself as a representation of the graph. The notion of convolutional networks for graphs as a generalization of classical convolutional networks for images was introduced by Bruna et al. (2013); Henaff et al. (2015). A key contribution here is the definition of graph convolution in the spectral domain using graph Fourier transform theory. Since then a number of works have focused on simplifying spectral convolutions to be localized (Defferrard et al., 2016) and easy to compute (Kipf & Welling, 2016). However spectral approaches do not generalize readily to different graphs due to their reliance on the particular Fourier basis they were trained on. To address this, spatial convolution methods are considered in Dai et al. (2017); Monti et al. (2016); Niepert et al. (2016); Such et al. (2017) for different applications. In Appendix A we discuss these models in detail where we show that the local spectral GCNNs and spatial GCNNs are mathematically equivalent, providing a unifying view of the variety of GCNN representations in the literature.

**Combinatorial optimization.** Starting with the work Hopfield & Tank (1985), performing combinatorial optimization using neural networks, and traveling salesman problem in particular, has been a topical subject in deep learning (Bello et al., 2016). More recently Vinyals et al. (2015); Bello et al. (2016) consider the traveling salesman problem using reinforcement learning. However the papers consider two-dimensional coordinates for vertices (e.g. cities on a map), without any explicit graph structure. As a more general solution Graves et al. (2016) proposes a differential neural computer

that is able to perform tasks like finding the shortest path. The work of Dai et al. (2017) is closest to ours in its empirical evaluation of spatial GCNN representation used with a reinforcement learning framework on combinatorial optimization problems. Our empirical results are significantly stronger, both in graph classes as well as graph sizes, than this very recent baseline.

## 3 GRAPHS AS DYNAMICAL SYSTEMS

### 3.1 GRAPH2SEQ

The central idea behind GRAPH2SEQ is that the trajectory of an appropriately chosen dynamical system induced by a graph, is a good representation for the graph. Such a representation has the advantage of progressively capturing more and more information about the graph as the trajectory unfolds.

Consider a directed graph $G(V, E)$ we seek to represent. Undirected graphs will be simply represented by having bi-directional edges between a pair of connected vertices. We create a discrete-time dynamical system in which vertex $v$ has a state of $\mathbf{x}_v(t) \in \mathbb{R}^d$ at time $t \in \mathbb{N}$, for all $v \in V$, and $d$ is the dimension of the state space. In GRAPH2SEQ, we consider an evolution rule of the form

$$\mathbf{x}_v(t+1) = g(\{\mathbf{x}_u(t) : u \in \Gamma(v)\}) + \mathbf{n}_v(t+1), \quad \forall v \in V, t \in \mathbb{N}, \tag{1}$$

where $g(\cdot)$ is a deterministic function that maps a set of vectors in $\mathbb{R}^d$ to another vector in $\mathbb{R}^d$, and $\mathbf{n}_v(\cdot)$ is a $d$-dimensional Gaussian circular noise (mean 0, covariance $I$). $u \in \Gamma(v)$ if there is a (directed) edge from $u$ to $v$ in the (directed) graph $G$. Specifically, in this paper we consider a transformation function $g(\cdot)$ that results in the following update rule:

$$\mathbf{x}_v(t+1) = \text{ReLU}(\mathbf{W}_0(\sum_{u \in \Gamma(v)} \mathbf{x}_u(t)) + \mathbf{b}_1) + \mathbf{n}_v(t+1), \quad \forall v \in V, t \in \mathbb{N}, \tag{2}$$

where $\mathbf{W}_0 \in \mathbb{R}^{d \times d}$ and $\mathbf{b} \in \mathbb{R}^{d \times 1}$ are trainable parameters. $\text{ReLU}(x) = \max(x, 0)$. Starting with an initial value (e.g., random or all zero) for the vertex state vectors $\mathbf{x}_v(0)$, Equation (2) above defines a dynamical system, the (random) trajectory of which is the GRAPH2SEQ representation. More generally, graphs could have features on their vertices or edges (e.g., weights on vertices) and they can be included in the evolution rule by appending them to the state vectors in a straight-forward way; these generalizations are outside the scope of this paper.

GRAPH2SEQ **= Seq2Graph**. Our first main result is that the representation of GRAPH2SEQ allows recovery of the adjacency matrix of the graph with arbitrarily high probability (here the randomness is with respect to the inherent randomness in GRAPH2SEQ due to the noise it adds). Specifically:

**Theorem 1.** *For any directed graph $G$ and associated (random) representation GRAPH2SEQ$(G)$ with sequence length $t$, there exists an inference procedure (with time complexity polynomial in $t$) that produces an estimate $\hat{G}_t$ such that $\lim_{t \to \infty} \mathbb{P}[G \neq \hat{G}_t] = 0$.*

**Importance of randomization**. Notice that GRAPH2SEQ's evolution rule (Equation (2)) includes a noise term that is added to the transformation function. The importance of randomization is highlighted by the following result (proof in Appendix B.2).

**Proposition 1.** *Under any deterministic evolution rule, there exists a graph $G$ which cannot be reconstructed exactly from its GRAPH2SEQ representation.*

The key point is that noise *breaks symmetry* in the (otherwise deterministic) dynamical system. Observe that vertices in the graph, besides any intrinsic features they may have, are not explicitly assigned unique identifiers. The deterministic evolution function, together with lack of identifying labels on vertices make distinguishing vertices difficult. The proof of Proposition 1 illustrates this for regular graphs.

### 3.2 FORMAL COMPUTATION MODEL

Although GRAPH2SEQ is an invertible representation of a graph, it is unclear how it compares to other GCNN representations in the literature. Below we define a formal computational model on

graphs, called LOCAL-GATHER, that includes GRAPH2SEQ as well as a large class of GCNN representations in the literature. Abstracting different representations into a formal computational model allows us reason about the fundamental limits of these methods. We show that fixed-depth GCNNs cannot fundamentally compute certain functions over graphs, where a sequence based representation such as GRAPH2SEQ is able to do so. For simplicity of notation we consider undirected graphs in this section and in the rest of this paper.

LOCAL-GATHER **Model**. Consider an undirected graph $G(V, E)$ (in which the vertices/edges can have some features associated with them) on which we seek to compute a function $f$. In the $k$-LOCAL-GATHER model, computations proceed in two rounds: In the *local step*, each vertex $v$ computes a representation $r(v)$ that depends only on the subgraph of vertices that are at a distance of at most $k$ from $v$. Following this, in the *gather step*, the function $f$ is computed by applying another function $g(\cdot)$ over the collection of aggregated representations $\{r(v) : v \in V\}$. We first note that GRAPH2SEQ is an instance of the $\infty$-LOCAL-GATHER model. Further, GCNNs using localized filters with a global aggregation (Kipf & Welling (2016); Dai et al. (2017), discussed in detail in Section A) fit this model (proof in Appendix B.3).

**Proposition 2.** *The spectral GCNN representation in Kipf & Welling (2016) and the spatial GCNN representation in Dai et al. (2017) belong to the* $4$-LOCAL-GATHER *model.*

**Fixed-length representations are insufficient**. We show below that for a fixed $k > 0$, no algorithm from the $k$-LOCAL-GATHER model can compute certain canonical graph functions exactly, with the proof relegated to Appendix B.4.

**Theorem 2.** *For any fixed $k > 0$, there exists a function $f(\cdot)$ and an input graph instance $G$ such that no $k$-LOCAL-GATHER algorithm can compute $f(G)$ exactly.*

For the graph and instance we have in Theorem 2, we present in Appendix B.5 a sequence-based representation (from the $\infty$-LOCAL-GATHER) that is able to asymptotically compute the function. Such a demonstration shows that sequence-based representations are strictly better than fixed-length graph representations in the LOCAL-GATHER model but also gives a sense for how sequential representations potentially operate in trained neural networks.

## 3.3 GEOMETRY OF GRAPH2SEQ DYNAMICS

Understanding how the graph structure is being encoded in the dynamics of Equation (2) involves answering questions such as: Does Equation (2) have a fixed point? Are there values for parameters $\mathbf{W}_0$ and $\mathbf{b}_1$ that recover some basic graph properties (e.g., the adjacency matrix)? While a full theoretical analysis of Equation (2) appears challenging partly due to the coupling of the $\mathbf{x}$ variables and the nonlinearity of the ReLU, we report the following empirical observations that at least partially help characterize the nature of the evolution.

We consider an experiment in which $d = 16$, and the entries of $\mathbf{W}_0, \mathbf{b}_1$ are drawn from an i.i.d. uniform over [0,1]. The state vector $\mathbf{x}_v$ is initialized to zero for all vertices. We consider graphs of size between 10–50 and of four types: random Erdos-Renyi, random tree, random regular graphs and random bipartite graph. In each case we perform the recursion of Equation (2) at least 10 times. The following general observations hold with high probability with respect to both parameter values, and the graph. Perhaps surprisingly, these observations also hold true on our model with trained parameters (e.g., for finding minimum vertex covers), as we discuss in Appendix D.

**(1) Convergence.** Depending on the variance of the initializing distribution for $\mathbf{W}_0$, the state vectors either converge to zero (small variance), or blow up (large variance).

**(2) Dimension collapse.** Roughly half the entries in the 16-dimensional state vector quickly become zeros (in about 4 or 5 iterations) and stay at zero. The indices of these zeroed-out entries are the same in state vectors across all the vertices.

**(3) Principal components' alignment.** We compute the principal component direction of the collection of vertex state vectors at each iteration, and observe that it converges. Figure 1(c) shows the absolute value of the inner product between the (normalized) principal component direction at each iteration with the principal component direction at iteration 10, for the Erdos-Renyi graph shown in Figure 1(a).

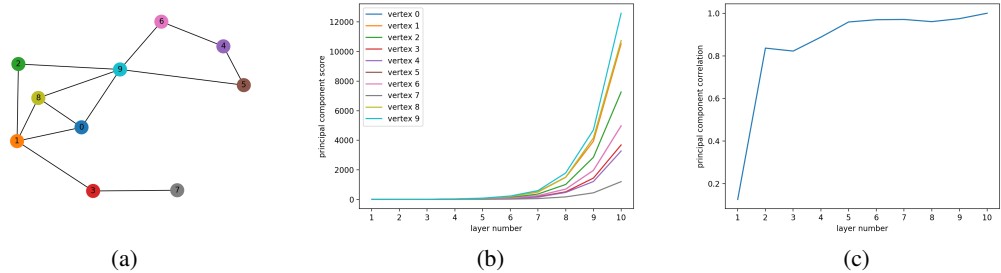

Figure 1: (a) Erdos-Renyi graph of size 10 considered in Figures (b) and (c), (b) vertex-wise principal component scores at each iteration, and (c) projection of the principal direction at each iteration on the principal direction of iteration 10. The vertex color and the line color in Figures (a) and (b) are matched.

**(4) Principal component scores and local connectivity.** In each iteration, we compute the principal score (projection along principal direction) of each vertex state vector. We observe that the principal score for a vertex roughly corresponds to its 'local connectivity' in the graph. The stronger a vertex is connected, the larger is the principal score. Fig. 1(b) shows this effect for the graph in Fig. 1(a) — vertex 9 (bright blue) is strongly connected and has the largest principal score, while vertex 7 (dark grey) has the least score.

## 4 NEURAL NETWORK DESIGN

We consider a reinforcement learning formulation for combinatorial optimization problems on graphs. Reinforcement learning is well-suited to such problems since the true 'labels' (i.e., the optimal solution) may be unavailable or hard to compute. Additionally, there are well-defined objective functions in most graph problems that can naturally be set as reward signals in the reinforcement learning model. Such an approach has been explored in recent works Vinyals et al. (2015); Bello et al. (2016); Dai et al. (2017) under different representation techniques. We use the GRAPH2SEQ representation in our learning setting which has key two features:

*Length of* GRAPH2SEQ *output*. The trajectories output by GRAPH2SEQ are subsequently fed to sequence processing units (specifically RNN-LSTM). Henceforth we call the combination of the GRAPH2SEQ representation and the subsequent sequence processing units together as the GRAPH2SEQ neural network architecture. The key feature of this design is that the length of the sequential representation is *not* fixed; it can vary depending on the input instance. We show that our model is able to learn rules – for both generating the sequence and processing it with the LSTM – that generalize to operate on arbitrarily long sequences. In turn, this translates to algorithmic solutions that can scale to large graph sizes.

*Adversarial training*. Another important reason for GRAPH2SEQ's generalization capability is our careful adversarial choice of the training set. The combinatorial problems naturally allow us to present 'hard examples' for training. By choosing an adversarial set of example graphs, we demonstrate the corresponding model generalizes much better both in terms of graph structure and graph size, compared to GCNN methods in the literature.

### 4.1 LEARNING ALGORITHM AND NETWORK ARCHITECTURE

**Reinforcement learning model.** We consider a reinforcement learning formulation in which vertices are chosen *one at a time*. Each time the agent chooses a vertex, it incurs a reward. The goal of training is to learn a policy such that cumulative reward is maximized. To achieve this goal, we use $Q$-learning to train the network. For input graph instance $G(V, E)$, a subset $S \subseteq V$ and $a \in V \backslash S$, this involves parametrically approximating a *Q-function* $Q(G, S, a)$. Here $S$ represents the set of vertices already picked. Appendix C.1 has the precise definitions of our reinforcement learning model and the $Q$-learning formulation. In the following we discuss the neural network to compute $Q(G, S, a)$.

**Network architecture.** The neural network comprises of three modules:

(1) *Graph2Seq*, that takes as input the graph $G$ and set $S$ of vertices chosen so far. It generates a sequence of vectors as output for each vertex.

(2) *Seq2Vec* reads the sequential representation produced by GRAPH2SEQ and summarizes it into a vector (for each vertex).

(3) *Q-Network* takes as input the vector summary of each vertex $a \in V$, and outputs the estimated $Q(G, S, a)$ value.

The overall architecture is illustrated in Fig. 2. To make the network practical, we truncate the sequence outputs of Graph2Seq to a length of $T$. However the value of $T$ is not fixed and will be varied both during training and testing, according to the size and complexity of the graph instances encountered. We discuss more on this in Section 4.2. For now let us suppose $T$ has a fixed value. We describe each module below.

**Graph2Seq.** We consider a $d$-dimensional state-space in which the dynamics of each vertex evolves. At time-step $t$, let $\mathbf{x}_v(t)$ denote the state of vertex $v$. Also, let $c_v(t)$ denote the binary variable that is one if $v \in S$ and zero otherwise. Then, the trajectory of each vertex $v \in V$ evolves according to

$$\mathbf{x}_v(t+1) = \text{ReLU}(\mathbf{W}_1 \sum_{u \in \Gamma(v)} \mathbf{x}_u(t) + \mathbf{w}_2 c_v(t) + \mathbf{b}_3), \tag{3}$$

for $t = 0, 1, \ldots, T-1$. Here $\mathbf{W}_1 \in \mathbb{R}^{d \times d}, \mathbf{w}_2 \in \mathbb{R}^d, \mathbf{b}_3 \in \mathbb{R}^d$ are trainable parameters. $\mathbf{x}_v(0)$ is initialized to all-zeros for each $v \in V$.

**Seq2Vec.** The sequence $(\{\mathbf{x}_v(t) : v \in V\})_{t=1}^T$ is processed by a gated recurrent network that sequentially reads $\mathbf{x}_v(\cdot)$ vectors at each time index for all $v \in V$. For time-step $t \in \{1, \ldots, T\}$, let $\mathbf{y}_v(t) \in \mathbb{R}^d$ be the $d$-dimensional cell state, $\mathbf{i}_v(t) \in \mathbb{R}^d$ be the cell input and $\mathbf{f}_v(t) \in (0, 1)^d$ be the forgetting gate, for each vertex $v \in V$. Each time-step a fresh input $\mathbf{i}_v(t)$ is computed based on the current states $\mathbf{x}_u(t)$ of $v$'s neighbors in $G$. The cell state is updated as a convex combination of the freshly computed inputs $\mathbf{i}_v(t)$ and the previous cell state $\mathbf{y}_v(t-1)$, where the weighting is done according to a forgetting value $\mathbf{f}_v(t)$ that is also computed based on the current vertex states. The update equations for the input vector, forgetting value and cell state are chosen as follows:

$$\mathbf{i}_v(t+1) = \text{ReLU}(\mathbf{W}_4 \sum_{u \in \Gamma(v)} \mathbf{x}_v(t) + \mathbf{w}_5 c_v(t) + \mathbf{b}_6) \tag{4}$$

$$\mathbf{f}_v(t+1) = \text{sigmoid}(\mathbf{W}_7 \sum_{u \in V} \mathbf{x}_u(t) + \mathbf{b}_8) \tag{5}$$

$$\mathbf{y}_v(t+1) = \mathbf{f}_v(t+1) \odot \mathbf{i}_v(t+1) + (\mathbf{1} - \mathbf{f}_v(t+1)) \odot \mathbf{y}_v(t), \tag{6}$$

where $\mathbf{W}_4, \mathbf{W}_7 \in \mathbb{R}^{d \times d}$ and $\mathbf{w}_5, \mathbf{b}_6, \mathbf{b}_8 \in \mathbb{R}^d$ are trainable parameters, $t = 0, 1, \ldots, T-1$, and $\mathbf{1}$ denotes the $d$-dimensional all-ones vector, and $\odot$ is element-wise multiplication. $\mathbf{y}_v(0)$ is initialized to all-zeros for every $v \in V$. The cell state at the final time-step $\mathbf{y}_v(T), v \in V$ is considered to be the desired vector summary of the GRAPH2SEQ sequence.

**Q-Network.** In this last step, the $\mathbf{y}_v(T)$ for each vertex $v$ is used to estimate the $Q$-values as

$$\tilde{Q}(G, S, v) = \mathbf{w}_9^T \text{ReLU}(\mathbf{W}_{10} \sum_{u \in V} \mathbf{y}_u(T)) + \mathbf{w}_{11}^T \text{ReLU}\left(\mathbf{W}_{12} \mathbf{y}_v(T)\right), \tag{7}$$

with $\mathbf{W}_{10}, \mathbf{W}_{12} \in \mathbb{R}^{d \times d}$ and $\mathbf{w}_9, \mathbf{w}_{11} \in \mathbb{R}^d$ being learnable parameters. Notice that every transformation function in the network leading up to Equation (7) is differentiable. This makes the whole network differentiable, allowing us to compute the (stochastic) gradients of the loss function in Equation (23) for learning. Next we describe the general training and testing techniques used in the evaluations.

## 4.2 TRAINING AND TESTING

**Training**. During training, the length of the GRAPH2SEQ representation is truncated to five observations (i.e., $T = 5$, see Fig. 2). We train on synthetically generated graphs of size 15 from a certain adversarially chosen graph family (we use different graph families for different optimization problems). We observe that training on a 'feature-rich' set of examples, can give a better performance across a range of different graph types during testing, compared to training and testing within the

same graph type. For example, in the case of minimum vertex cover, training on (mismatched) 'planted vertex cover' examples (Section 5.1) has a much better generalization performance on Erdos-Renyi graphs, than if the same heuristic had been trained on Erdos-Renyi graphs.

In each case, our model is trained for 100,000 iterations, though we found convergence to occur typically much earlier. We use experience replay in which during each iteration we sample a random (state, action, next state) tuple from that was made previously, and use that to compute the gradient update. We use a learning rate of $10^{-3}$ with the Adam optimizer (Kingma & Ba, 2014), and an exploration probability that is reduced from $1.0$ initially to a resting value of $0.05$ over $10,000$ iterations. The amount of noise added in the evolution ($n_v(t)$ in Equation 2) seemed to not matter; we have set the noise variance to zero in all our experiments (training and testing).

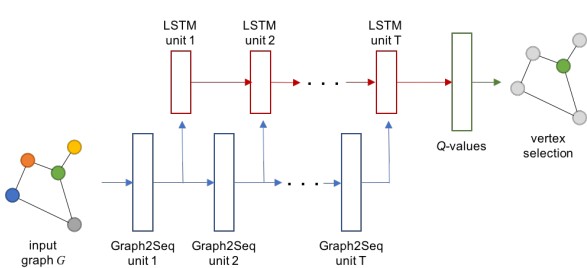

Figure 2: GRAPH2SEQ neural network architecture.

**Testing**. Due to the recurrent nature of GRAPH2SEQ (and the LSTM units), we can use the trained parameters to generate, and subsequently summarize, a sequential representation that is arbitrarily long. For any fixed $T > 0$, let GRAPH2SEQ($T$) denote the architecture obtained by restricting the sequence length to $T$ (Fig. 2). We run GRAPH2SEQ($T$) for every $T \in \mathbb{N}$ over an interval $[1, T_{\max}]$, where $T_{\max}$ is a hyper-parameter (we choose $T_{\max} < 40$ in our experiments). Notice for each fixed $T$ and input instance $G(V, E)$, GRAPH2SEQ($T$) outputs a solution set $S_T \subseteq V$. Since our goal is to maximize objective, we select the output of that $T \in [1, T_{\max}]$ that has the largest objective function value as our final output. This procedure is summarized in detail in Algorithm 2 in Appendix C.

We test generalization in graph structure and size. Using the *same* trained parameters, we test GRAPH2SEQ on the following broad graph classes: Erdos-Renyi graphs, bipartite graphs, regular graphs and structured examples, while varying the graph sizes between 25–3200.

## 5 EVALUATION: MINIMUM VERTEX COVER

In this section we present evaluation results for the minimum vertex cover (MVC). Results for the max cut and maximum independent set problems are in Appendix E.

The MVC of a graph $G(V, E)$ is a set $S \subseteq V$ of the lowest cardinality such that for every edge $(u, v) \in E$ at least one of $u$ or $v$ are in $S$. Approximation algorithms to within a factor 2 are known for MVC, however it cannot be approximated better than 1.3606 unless $P = NP$. To model this problem using RL, we define the set of vertices chosen as actions across iterations to be our estimated vertex cover output. Each time a vertex $v$ is chosen, the learning agent receives a reward of $-1$ and we set $c_v \leftarrow 1$ for all subsequent iterations. During each iteration we pick as action the vertex with the highest $Q$-value that has not be chosen already (i.e., for which $c_v$ is 0). The algorithm terminates when there is no more edge to be covered. Throughout our evaluations we have used state vector dimension $d = 16$, although the empirical findings stay roughly the same for $d \geq 8$.

### 5.1 ADVERSARIAL TRAINING

We train our model on *planted vertex-cover* graph examples, in which a small graph is embedded within a larger graph such that the vertices of the smaller graph constitute the optimal minimum vertex cover. Such planted examples form a reasonably hard class of examples for MVC, and appears essential to the superior generalization of our trained model. When trained on other graph families (such as Erdos-Renyi graphs), the generalization (both with respect to size, and graph structure including Erdos-Renyi itself) is much less pronounced than training with the planted examples.

To generate our examples, we first sample a random Erdos-Renyi graph $G_i$ (to be planted) of size 5 and edge probability $p = 0.15$. Next, we define a second graph $G_o$ of size 10 in which all the vertices are disjoint. The overall graph is formed by connecting each vertex of $G_o$ to each vertex

of $G_i$. Note that the vertices of $G_i$ form the optimum vertex cover in these examples. With these training examples, we follow the general training strategy described in Section 4.2.

## 5.2 TESTING

We test generalization capability along two dimensions: graph structure and graph size.

**Graph types and sizes.** We consider graphs of the following four types:

(1) Erdos-Renyi graph, with edge probability $p = 0.15$.

(2) Random regular graph, with degree $4$.

(3) Random bipartite graph, both partites of equal size and edge probability $p = 0.75$.

(4) Worst-case examples for Greedy. The Greedy algorithm is a well-known MVC heuristic in which we sequentially select the vertex having the largest number of uncovered edges remaining in the graph. This algorithm can be shown to have a $O(\log n)$ approximation for vertex cover, by means of constructive examples where they perform poorly (Johnson, 1973). One popular construction is a bipartite graph with $n$ vertices on one side (say, partite $P_1$), and $\sum_{i=2}^{n} \lfloor n/i \rfloor$ vertices on the other partite ($P_2$). For $i = 2, \ldots, n$, $\lfloor n/i \rfloor$ vertices from $P_2$ have a degree of $i$ and each vertex from $P_1$ is connected to at most one vertex of degree $i$.

For each type, we test on graphs of size ranging from 25–3200 in exponential increments. Crucially, we use the *same trained model* on all of the test examples. We also limit the number of layers in GRAPH2SEQ (i.e., the sequence length) to 15 in our evaluations.

**Baseline**. We compare our results against the following baseline algorithms:

(1) Greedy algorithm for vertex cover (described above).

(2) Fixed-depth GCNN trained on Erdos-Renyi graphs. We consider a 5-layer GCNN in which only the outputs of the 5-th layer are fed to the $Q$-learning network. This algorithm is identical to Dai et al. (2017). This network is trained on size-15 Erdos-Renyi graphs with $p = 0.15$.

(3) Fixed-depth GCNN trained on planted vertex-cover graphs. This baseline is the same as above, but trained on the adversarial planted-cover examples of Section 5.1.

(4) Matching heuristic, a 2-approximation algorithm that selects an arbitrary edge in each round, and includes both end-points in the cover. The selected vertices are then removed from the graph and the process repeats.

(5) List heuristic. We also compare against the algorithm proposed recently in Shimizu et al. (2016) that outperforms previous vertex cover heuristics.

The ground truth is found via the Gurobi optimization package which solves the MVC integer program with a time cut-off of 240s. The solver found the optimal solution for all graphs up to size 200 before this time. We report our results via the approximation ratio of the values returned by the baseline algorithms considered to the ground truth found by the Gurobi solver.

**Results.** Figure 3 plots results for the different baselines under each of four graph types considered in our experiment and we make the following observations: (a) In general, we see that GRAPH2SEQ has a performance that is consistently within 5% of the optimal (or the time-limited IP solution where applicable), across the range of graph types and sizes considered. The other baselines, however, demonstrate behavior that is not consistent and have certain classes of graph types and/or sizes where they perform poorly. (b) The Greedy baseline has a near-optimal performance, comparable to GRAPH2SEQ, with Erdos-Renyi and random bipartite graphs. It is slightly worse than GRAPH2SEQ on random regular graphs, and does very poorly (not surprisingly) on the worst-case examples. (c) The fixed-depth GCNN baselines behave remarkably differently from each other. The one trained on Erdos-Renyi graphs, i.e., Dai et al. (2017), has a performance comparable to GRAPH2SEQ and Greedy with Erdos-Renyi and random regular graphs. However with random bipartite graphs it does poorly at graph sizes 200 or less. Similarly, with the worst-case example, the heuristic does not do well at graph sizes 50 or above. (d) The fixed-depth GCNN baseline trained on planted-vertex cover examples demonstrates poor approximation ratio in all graph categories. (e) Similarly the matching heuristic is also consistently poor in all categories. (f) The list heuristic of Shimizu et al. (2016) has a comparable performance to GRAPH2SEQ in most cases. However in large random regular graphs, we observe its performance degrading 10% above optimal.

**Geometry and semantics of encoding**. Towards an understanding of what aspect of solving the MVC is learnt, we conduct empirical studies on the dynamics of the state vectors as well as present techniques and semantic interpretations of GRAPH2SEQ. First we observe that the same obser-

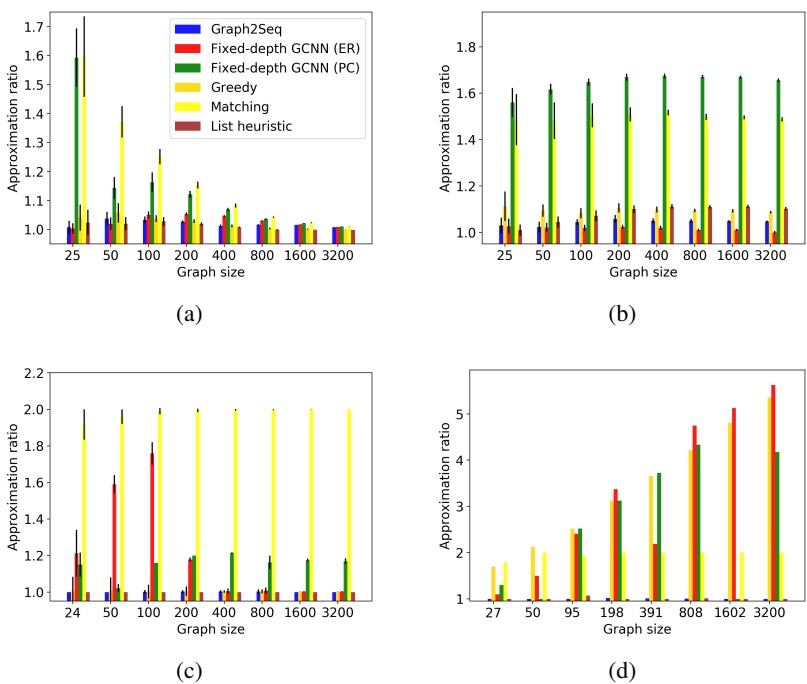

Figure 3: Minimum vertex cover in (a) random Erdos-Renyi graphs, (b) random regular graphs, (c) random bipartite graphs, (d) greedy example. The error bars show one standard deviation.

vations (cf. Section 3.3) as for random parameter choices continue to hold. The most interesting semantics are observed in the $Q(\cdot)$ function which has two components (cf. Equation (7)). The first term, denoted by $Q_1$, is the same for all the vertices and includes a sum of all the $\mathbf{y}(\cdot)$ vectors. The second term, denoted by $Q_2(v)$ depends on the $\mathbf{y}(\cdot)$ vector for the vertex being considered.

We consider these two values at the end of the first round of the learning algorithm (with $S = \{\}$) for a planted vertex cover graph of size 15. We make two observations: (a) the values of $Q_1$ and $Q_2(\cdot)$ are close to being integers. $Q_1$ has a value that is one less than the negative of the minimum vertex cover size. (b) For a vertex $v$, $Q_2(v)$ is binary valued from the set $\{0, 1\}$. $Q_2(v)$ is one, if vertex $v$ is part of an optimum vertex cover, and zero otherwise. Thus the neural network, in principle, computes the complete set of vertices in the optimum cover at the very first round itself. A detailed description of these semantics and a visualization of the various phenomena is in Appendix D.

## 6 CONCLUSION

We have proposed GRAPH2SEQ that represents graphs as infinite time-series of vectors, one for each vertex of the graph. The time-series representation melds naturally with modern RNN architectures that take time-series as inputs. We have demonstrated the strong synergistic benefits towards solving three canonical combinatorial optimization problems on undirected graphs, ranging across the complexity-theoretic hardness spectrum. Our empirical results best state-of-the-art approximation algorithms for these problems on a variety of graph sizes and types. In particular, GRAPH2SEQ is strictly better in generalization capabilities than deep learning techniques in the literature (GCNN methods). An open direction involves a more systematic study of the capabilities of GRAPH2SEQ across the panoply of graph combinatorial optimization problems, as well as its performance in concrete (and myriad) downstream applications. Another open direction involves interpreting key principles learnt by the neural network in solving any specific combinatorial optimization problem. Traditional understanding of heuristic algorithms involves bounding their worst-case approximation ratio, which appears impossible to conduct in the case of neural network methods – exploring modern interpretability methods in machine learning (example: LIME Ribeiro et al. (2016)) is of interest.

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

## A    Background: Graph Convolutional Neural Networks

An ideal graph representation is one that captures all innate structures of the graph relevant to the task at hand, and moreover can also be learned via gradient descent methods. However, this is challenging since the relevant structures could range anywhere from local attributes (example: node degrees) to long-range dependencies spanning across a large portion of the graph (example: does there exist a path between two vertices) (Kuhn et al., 2016). Such broad scale variation is also a well-known issue in computer vision (image classification, segmentation etc.), wherein convolutional neural network (CNN) designs have been used quite successfully (Krizhevsky et al., 2012). Perhaps motivated by this success, recent research has focused on generalizing the traditional CNN architecture to develop designs for graph convolutional neural networks (GCNN) (Bruna et al., 2013; Niepert et al., 2016). By likening the relationship between adjacent pixels of an image to that of adjacent nodes in a graph, the GCNN seeks to emulate CNNs by defining localized 'filters' with shared parameters.

Current GCNN filter designs can be classified into one of two categories: *spatial* (Kipf & Welling, 2016; Dai et al., 2017; Nowak et al., 2017), and *spectral* (Defferrard et al., 2016). For an integral hyper-parameter $K \geq 0$, filters in either category process information from a $K$-local neighborhood surrounding a node to compute the output. Here we consider localized spectral filters such as proposed in Defferrard et al. (2016). The difference between the spatial and spectral versions arises in the precise way in which the aggregated local information is combined.

*Spatial GCNN.* For input feature vector $\mathbf{x}_v$ at each node $v \in V$ of a graph $G$, a spatial filtering operation is the following:

$$\mathbf{y}_v = \sigma \left( \sum_{k=0}^{K-1} \mathbf{W}_k \left( \sum_{u \in V} (\tilde{\mathbf{A}}^k)_{v,u} \mathbf{x}_u \right) + \mathbf{b}_0 \right) \quad \forall v \in V, \tag{8}$$

where $\mathbf{y}_v$ is the filter output, $\mathbf{W}_k, k = 1, \ldots, K$ and $\mathbf{b}_0$ are learnable parameters, and $\sigma$ is a non-linear activation function that is applied element-wise. $\tilde{\mathbf{A}} = \mathbf{D}^{-1/2} \mathbf{A} \mathbf{D}^{-1/2}$ is the normalized adjacency matrix, and $\mathbf{D}$ is the diagonal matrix of vertex degrees. Use of un-normalized adjacency matrix is also common. The $k$-power of the adjacency matrix selects nodes a distance of at most $k$ hops from $v$. ReLU is a common choice for $\sigma$. We highlight two aspects of spatial GCNNs: (i) the feature vectors are aggregated from neighboring nodes directly specified through the graph topology, and (ii) the aggregated features are summarized via an addition operation.

*Spectral GCNN.* Spectral GCNNs use the notion of graph Fourier transforms to define convolution operation as the inverse transform of multiplicative filtering in the Fourier domain. Since this is a non-local operation potentially involving data across the entire graph, and moreover it is computationally expensive to compute the transforms, recent work has focused on approximations to produce a local spectral filter of the form

$$\mathbf{y}_v = \sigma \left( \sum_{k=0}^{K-1} \mathbf{W}'_k \left( \sum_{u \in V} (\tilde{\mathbf{L}}^k)_{v,u} \mathbf{x}_u \right) + \mathbf{b}'_0 \right) \quad \forall v \in V, \tag{9}$$

where $\tilde{\mathbf{L}} = \mathbf{I} - \tilde{\mathbf{A}}$ is the normalized Laplacian of the graph, $(\tilde{\mathbf{L}}^k)_{v,u}$ denotes the entry at the row corresponding to vertex $v$ and column corresponding to vertex $u$ in $\tilde{\mathbf{L}}^k$, and $\mathbf{W}'_k, \mathbf{b}'_0$ are parameters (Defferrard et al., 2016; Kipf & Welling, 2016). As in the spatial case, definitions using un-normalized version of Laplacian matrix are also used. $\sigma$ is typically the identity function here. Equation (9) is a local operation because the $k$-th power of the Laplacian, at any row $v$, has a support no larger than the $k$-hop neighborhood of $v$. Thus, while the aggregation is still localized, the feature vectors are now weighted by the entries of the Laplacian before summation.

*Spectral and Spatial GCNN are equivalent.* The distinction between spatial and spectral convolution designs is typically made owing to their seemingly different definitions. However we show that both designs are mathematically equivalent in terms of their representation capabilities.

**Proposition 3.** *Consider spatial and spectral filters in Equations* (8) *and* (9)*, using the same non-linear activation function $\sigma$ and $K$. Then, for graph $G(V, E)$, for any choice of parameters $\mathbf{W}_k$ and $\mathbf{b}_0$ for $k = 1, \ldots, K$ there exists parameters $\mathbf{W}'_k$ and $\mathbf{b}'_0$ such that the filters represent the same transformation function, and vice-versa.*

*Proof.* Consider a vertex set $V = \{1, 2, \ldots, n\}$ and $d$-dimensional vertex states $\mathbf{x}_i$ and $\mathbf{y}_i$ at vertex $i \in V$. Let $\mathbf{X} = [\mathbf{x}_1, \ldots, \mathbf{x}_n]$ and $\mathbf{Y} = [\mathbf{y}_1, \ldots, \mathbf{y}_n]$ be the matrices obtained by concatenating the state vectors of all vertices. Then the spatial transformation function of Equation (8) can be written as

$$\mathbf{Y} = \sigma \left( \sum_{k=0}^{K-1} \mathbf{W}_k \mathbf{X} \tilde{\mathbf{A}}^k + \mathbf{b}_0 \mathbf{1}^T \right), \tag{10}$$

while the spectral transformation function of Equation (9) can be written as

$$\mathbf{Y} = \sigma \left( \sum_{k=0}^{K-1} \mathbf{W}'_k \mathbf{X} \tilde{\mathbf{L}}^k + \mathbf{b}'_0 \mathbf{1}^T \right) \tag{11}$$

$$= \sigma \left( \sum_{k=0}^{K-1} \mathbf{W}'_k \mathbf{X} (\mathbf{I} - \tilde{\mathbf{A}})^k + \mathbf{b}'_0 \mathbf{1}^T \right) \tag{12}$$

$$= \sigma \left( \sum_{k=0}^{K-1} \mathbf{W}'_k \mathbf{X} \sum_{i=0}^{k} \binom{k}{i} (-1)^{k-i} \tilde{\mathbf{A}}^i + \mathbf{b}'_0 \mathbf{1}^T \right) \tag{13}$$

$$= \sigma \left( \sum_{k=0}^{K-1} \left( \sum_{i=k}^{K-1} \mathbf{W}'_i \binom{i}{k} (-1)^{i-k} \right) \mathbf{X} \tilde{\mathbf{A}}^k + \mathbf{b}'_0 \mathbf{1}^T \right). \tag{14}$$

Equation (12) follows by the definition of the normalized Laplacian matrix, and Equation (13) derives from binomial expansion. To make the transformation in Equations (10) and (14) equal, we can set

$$\sum_{i=k}^{K-1} \mathbf{W}'_i \binom{i}{k} (-1)^{i-k} = \mathbf{W}_k, \quad \forall\, 0 \le k \le K - 1, \tag{15}$$

and check if there are any feasible solutions for the primed quantities. Clearly there are, with one possible solution being $\mathbf{b}'_0 = \mathbf{b}_0$ and

$$\mathbf{W}'_{K-1} = \mathbf{W}_{K-1} \tag{16}$$

$$\mathbf{W}'_k = \mathbf{W}_k - \sum_{i=k+1}^{K-1} \mathbf{W}'_i \binom{i}{k} (-1)^{i-k}, \quad \forall\, 0 \le k \le K - 2. \tag{17}$$

Thus for any choice of values for $\mathbf{W}_k, \mathbf{b}_0$ for $k = 0, \ldots, K - 1$ there exists $\mathbf{W}'_k, \mathbf{b}'_0$ for $k = 0, \ldots, K - 1$ such that the spatial and spectral transformation functions are equivalent. The other direction (when $\mathbf{W}'_k$ and $\mathbf{b}_0$ are fixed), is similar and straightforward. $\qquad \square$

Depending on the application, the convolutional layers may be supplemented with pooling and coarsening layers that summarize outputs of nearby convolutional filters to form a progressively more compact spatial representation of the graph. This is useful in classification tasks where the desired output is one out of a few possible classes (Bruna et al., 2013). For applications requiring decisions at a per-node level (e.g. community detection), a popular strategy is to have multiple repeated convolutional layers that compute vector representations for each node, which are then processed to make a decision (Dai et al., 2017; Bruna & Li, 2017; Nowak et al., 2017). The conventional wisdom here is to have as many layers as the diameter of the graph, since filters at each layer aggregate information only from nearby nodes. Such a strategy is sometimes compared to the message passing algorithm (Gilmer et al., 2017), though the formal connections are not clear as noted in Nowak et al. (2017). Finally the GCNNs described so far are all end-to-end differentiable and can be trained using mainstream techniques for supervised, semi-supervised or reinforcement learning applications.

Other lines of work use ideas inspired from word embeddings for graph representation (Grover & Leskovec, 2016; Perozzi et al., 2014). Post-GCNN representation, LSTM-RNNs have been used to analyze time-series data structured over a graph. Seo et al. (2016) propose a model which combines GCNN and RNN to predict moving MNIST data. Liang et al. (2016) design a graph LSTM for semantic object parsing in images.

# B    Section 3 Proofs

## B.1    Proof of Theorem 1

*Proof.* Consider a GRAPH2SEQ trajectory on graph $G(V, E)$ according to Equation (2) in which the vertex states are initialized randomly from some distribution. Let $\mathbf{X}_v(t)$ (resp. $\mathbf{x}_v(t)$) denote the random variable (resp. realization) corresponding to the state of vertex $v$ at time $t$. For time $T > 0$ and a set $S \subseteq V$, let $\mathbf{X}_S^T$ denote the collection of random variables $\{\mathbf{X}_v(t) : v \in S, 0 \leq t \leq T\}$; $\mathbf{x}_V^T$ will denote the realizations.

An information theoretic estimator to output the graph structure by looking at the trajectory $\mathbf{X}_V^T$ is the directed information graph considered in Quinn et al. (2015). Roughly speaking, the estimator evaluates the conditional directed information for every pair of vertices $u, v \in V$, and declares an edge only if it is positive (see Definition 3.4 in Quinn et al. (2015) for details). Estimating conditional directed information efficiently from samples is itself an active area of research Quinn et al. (2011), but simple plug-in estimators with a standard kernel density estimator will be consistent. Since the theorem statement did not specify sample efficiency (i.e., how far down the trajectory do we have to go before estimating the graph with a required probability), the inference algorithm is simple and polynomial in the length of the trajectory. The key question is whether the directed information graph is indeed the same as the underlying graph $G$. Under some conditions on the graph dynamics (discussed below in Properties 1–3), this holds and it suffices for us to show that the dynamics generated according to Equation (2) satisfies those conditions.

**Property 1.** *For any $T > 0$, $P_{\mathbf{X}_V^T}(\mathbf{x}_V^T) > 0$ for all $\mathbf{x}_V^T$.*

This is a technical condition that is required to avoid degeneracies that may arise in deterministic systems. Clearly GRAPH2SEQ's dynamics satisfies this property due to the additive i.i.d. noise in the transformation functions.

**Property 2.** *The dynamics is strictly causal, that is $P_{\mathbf{X}_V^T}(\mathbf{x}_V^T)$ factorizes as* $\prod_{t=0}^{T} \prod_{v \in V} P_{\mathbf{X}_v(t)|\mathbf{X}_V^{t-1}}(\mathbf{x}_v(t)|\mathbf{x}_V^{t-1})$.

This is another technical condition that is readily seen to be true for GRAPH2SEQ. The proof also follows from Lemma 3.1 in Quinn et al. (2015).

**Property 3.** *$G$ is the minimal generative model graph for the random processes $\mathbf{X}_v(t), v \in V$.*

Notice that the transformation operation Equation (2) in our graph causes $\mathbf{X}_V^T$ to factorize as

$$P_{\mathbf{X}_V^T}(\mathbf{x}_V^T) = \prod_{t=0}^{T} \prod_{v \in V} P_{\mathbf{X}_v(t)|\mathbf{X}_{\Gamma(v)}^{t-1}}(\mathbf{x}_v(t)|\mathbf{x}_{\Gamma(v)}^{t-1}) \tag{18}$$

for any $T > 0$, where $\Gamma(v)$ is the set of neighboring vertices of $v$ in $G$. Now consider any other graph $G'(V, E')$. $G'$ will be called a minimal generative model for the random processes $\{\mathbf{X}_v(t) : v \in V, t \geq 0\}$ if
(1) there exists an alternative factorization of $P_{\mathbf{X}_V^T}(\mathbf{x}_V^T)$ as

$$P_{\mathbf{X}_V^T}(\mathbf{x}_V^T) = \prod_{t=0}^{T} \prod_{v \in V} P_{\mathbf{X}_v(t)|\mathbf{X}_{\Gamma'(v)}^{t-1}}(\mathbf{x}_v(t)|\mathbf{x}_{\Gamma'(v)}^{t-1}) \tag{19}$$

for any $T > 0$, where $\Gamma'(v)$ is the set of neighbors of $v$ in $G'$, and
(2) there does not exist any other graph $G''(V, E'')$ with $E'' \subset E$ and a factorization of $P_{\mathbf{X}_V^T}(\mathbf{x}_V^T)$ as $\prod_{t=0}^{T} \prod_{v \in V} P_{\mathbf{X}_v(t)|\mathbf{X}_{\Gamma''(v)}^{t-1}}(\mathbf{x}_v(t)|\mathbf{x}_{\Gamma''(v)}^{t-1})$ for any $T > 0$, where $\Gamma''(v)$ is the set of neighbors of $v$ in $G''$.

Intuitively, a minimal generative model is the smallest spanning graph that can generate the observed dynamics. To show that $G(V, E)$ is indeed a minimal generative model, let us suppose the contrary and assume there exists another graph $G'(V, E')$ with $E' \subset E$ and a factorization of $P_{\mathbf{X}_V^T}(\mathbf{x}_V^T)$ as

in Equation (19). In particular, let $v$ be any node such that $\Gamma'(v) \subset \Gamma(v)$. Then by marginalizing the right hand sides of Equations (18) and (19), we get

$$P_{\mathbf{X}_v(1)|\mathbf{X}^0_{\Gamma(v)}}(\mathbf{x}_v(1)|\mathbf{x}^0_{\Gamma(v)}) = P_{\mathbf{X}_v(1)|\mathbf{X}^0_{\Gamma'(v)}}(\mathbf{x}_v(1)|\mathbf{x}^0_{\Gamma'(v)}). \tag{20}$$

Note that Equation (20) needs to hold for all possible realizations of the random variables $\mathbf{X}_v(1), \mathbf{X}^0_{\Gamma(v)}$ and $\mathbf{X}^0_{\Gamma'(0)}$. However if the parameters $\mathbf{\Theta}_0$ and $\Theta_1$ in Equation (2) are generic, this is clearly not true. To see this, let $u \in \Gamma(v) \backslash \Gamma'(v)$ be any vertex. By fixing the values of $\mathbf{x}_v(1), \mathbf{x}^0_{\Gamma(v) \backslash \{u\}}$ it is possible to find two values for $\mathbf{x}_u(0)$, say $\mathbf{a_1}$ and $\mathbf{a_2}$, such that

$$\text{ReLU}\left(\mathbf{\Theta}_0\left(\sum_{i \in \Gamma(v)\backslash\{u\}} \mathbf{x}_i(0) + \mathbf{a}_1\right) + \Theta_1\right) \neq \text{ReLU}\left(\mathbf{\Theta}_0\left(\sum_{i \in \Gamma(v)\backslash\{u\}} \mathbf{x}_i(0) + \mathbf{a}_2\right) + \Theta_1\right). \tag{21}$$

As such the Gaussian distributions in these two cases will have different means. However the right hand side Equation (20) does not depend on $\mathbf{x}_u(0)$ at all, resulting in a contradiction. Thus $G$ is a minimal generating function of $\{\mathbf{X}_v(t) : v \in V, t \geq 0\}$. Thus Property 3 holds as well. Now the result follows from the following Theorem.

**Theorem 3** (Theorem 3.6, Quinn et al. (2015)). *If Properties 1, 2 and 3 are satisfied, then the directed information graph is equivalent to the graph $G$.*

□

### B.2 Proof of Proposition 1

*Proof.* Consider 4-regular graphs $R_1$ and $R_2$ with vertices $\{0, 1, \ldots, 7\}$ and edges $\{(0,3), (0,5), (0,6), (0,7), (1,2), (1,4), (1,6), (1,7), (2,3), (2,5), (2,6), (3,4), (3,5), (4,5),$ $(4,7), (6,7)\}$ and $\{(0,1), (0,2), (0,4), (0,7), (1,4), (1,5), (1,6), (2,3), (2,4), (2,7), (3,5), (3,6),$ $(3,7), (4,6), (5,6), (5,7)\}$ respectively. Then under a deterministic evolution rule, since $R_1$ and $R_2$ are 4-regular graphs, the trajectory will be identical at all nodes across the two graphs. However the graphs $R_1$ and $R_2$ are structurally different. For e.g., $R_1$ has a minimum vertex cover size of 5, while for $R_2$ it is 6. As such, if any one of the graphs ($R_1$, say) is provided as input to be represented, then from the representation it is impossible to exaclty recover $R_1$'s structure. □

### B.3 Proof of Proposition 2

*Proof.* Kipf & Welling (2016) use a two layer graph convolutional network, in which each layer uses convolutional filters that aggregate information from the immediate neighborhood of the vertices. This corresponds to a 2-local representation function $r(\cdot)$ in our computational model. Following this step, the values at the vertices are aggregated using softmax to compute a probability score at each vertex. Since this procedure is independent of the structure of the input graph, it is a valid gathering function $g(\cdot)$ in LOCAL-GATHER and the overall architecture belongs to a 2-LOCAL-GATHER model.

Similarly, Dai et al. (2017) also consider convolutional layers in which the neurons have a spatial locality of one. Four such convolutional layers are cascaded together, the outputs of which are then processed by a separate $Q$-learning network. Such a neural architecture is an instance of the 4-LOCAL-GATHER model. □

### B.4 Proof of Theorem 2

*Proof.* Consider a family $\mathcal{G}$ of undirected, unweights graphs. Let $f : \mathcal{G} \to \mathbb{Z}$ denote a function that computes the size of the minimum vertex cover of graphs from $\mathcal{G}$. For $k > 0$ fixed, let ALG denote any algorithm from the $k$-LOCAL-GATHER model, with a representation function $r_{\texttt{ALG}}(\cdot)$ and aggregating function $g_{\texttt{ALG}}(\cdot)$.[1] We present two graphs $G_1$ and $G_2$ such that $f(G_1) \neq f(G_2)$, but the set of computed states $\{r_{\texttt{ALG}}(v) : v \in G_i\}$ is the same for both the graphs ($i = 1, 2$). Now, since the

---

[1]See beginning of Section 3 for explanations of $r(\cdot)$ and $g(\cdot)$.

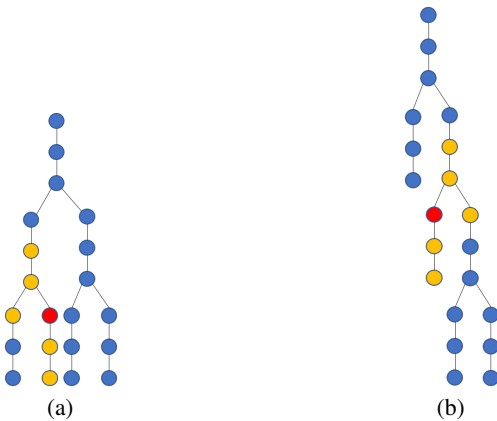

Figure 4: Example to illustrate $k$-LOCAL-GATHER algorithms are insufficient for computing certain functions. Corresponding vertices in the two trees above have similar local neighborhoods, but the trees have minimum vertex cover of different sizes.

gather function $g_{\text{ALG}}(\cdot)$ operates only on the set of computed states (by definition of our model), this implies ALG cannot distinguish between $f(G_1)$ and $f(G_2)$, thus proving our claim.

For simplicity, we fix $k = 2$ (the example easily generalizes for larger $k$). We consider the graphs $G_1$ and $G_2$ as shown in Fig. 4(a) and 4(b) respectively. To construct these graphs, we first consider binary trees $B_1$ and $B_2$ each having 7 nodes. $B_1$ is a completely balanced binary tree with a depth of 2, whereas $B_2$ is a completely imbalanced binary tree with a depth of 3. Now, to get $G_1$ and $G_2$, we replace each node in $B_1$ and $B_2$ by a chain of 3 nodes (more generally, by an odd number of nodes larger than $k$). At each location in $B_i$ ($i = 1, 2$), the head of the chain of nodes connects to the tail of the parent's chain of nodes, as shown in Fig. 4.

The sizes of the minimum vertex cover of $G_1$ and $G_2$ are 9 and 10 respectively. However, there exists a one-to-one mapping between the vertices of $G_1$ and the vertices of $G_2$ such that the 2-hop neighborhood around corresponding vertices in $G_1$ and $G_2$ are the same. For example, in Fig. 4(a) and 4(b) the pair of nodes shaded in red have an identical 2-hop neighborhood (shaded in yellow). As such, the representation function $r_{\text{ALG}}(\cdot)$ – which for any node depends only on its $k$-hop neighborhood – will be the same for corresponding pairs of nodes in $G_1$ and $G_2$.

Finally, the precise mapping between pairs of nodes in $G_1$ and $G_2$ is obtained as follows. First consider a simple mapping between pairs of nodes in $B_1$ and $B_2$ in which (i) the 4 leaf nodes in $B_1$ are mapped to the leaf nodes in $B_2$, (ii) the root of $B_1$ is mapped to the root of $B_2$ and (iii) the 2 interior nodes of $B_1$ are mapped to the interior nodes of $B_2$. We generalize this mapping to $G_1$ and $G_2$ in two steps: (1) mapping chains of nodes in $G_1$ to chains of nodes in $G_2$, according to the $B_1 - B_2$ map, and (2) within corresponding chains of nodes, we map nodes according to order (head-to-head, tail-to-tail, etc.). $\qquad\square$

### B.5 SEQUENTIAL HEURISTIC TO COMPUTE MVC ON TREES

Consider any unweighted, undirected tree $T$. We let the state at any node $v \in T$ be represented by a two-dimensional vector $[x_v, y_v]$. For any $v \in T$, $x_v$ takes values over the set $\{-\epsilon, +1\}$ while $y_v$ is in $\{-1, 0, \epsilon\}$. Here $\epsilon$ is a parameter that we choose to be less than one over the maximum degree of the graph. Semantically $x_v$ stands for whether vertex $v$ is 'active' ($x_v = +1$) or 'inactive' ($x_v = -\epsilon$). Similarly $y_v$ stands for whether $v$ has been selected to be part of the vertex cover ($y_v = +\epsilon$), has not been selected to be part of the cover ($y_v = -1$), or a decision has not yet been made ($y_v = 0$). Initially $x_v = -\epsilon$ and $y_v = 0$ for all vertices. The heuristic proceeds in rounds, wherein at each round any vertex $v$ updates its state $[x_v, y_v]$ based on the state of its neighbors as shown in Algorithm 1.

The update rules at vertex $v$ are (1) if $v$ is a leaf or if at least one of $v$'s neighbors are active, then $v$ becomes active; (2) if $v$ is active, and if at least one of $v$'s active neighbors have not been chosen in

---

**Algorithm 1:** Sequential heuristic to compute minimum vertex cover on a tree.

---

**Input:** Undirected, unweighted tree $T$; Number of rounds `NumRounds`
**Output:** Size of minimum vertex cover on $T$
$x_v(0) \leftarrow -\epsilon$ for all $v \in T$        `// ` $x_v(i)$ ` is ` $x_v$ ` at round ` $i$
$y_v(0) \leftarrow 0$ for all $v \in T$        `// ` $y_v(i)$ ` is ` $y_v$ ` at round ` $i$
`/* Computing the representation ` $r(v)$ ` for each ` $v \in T$    `*/`
**for** *i from 1 to* `NumRounds` **do**
     At each vertex $v$:
     **if** $\sum_{u \in \Gamma(v)} x_u(i-1) \geq -\epsilon$ **then**
         $x_v(i) \leftarrow +1$
         **if** $\sum_{u \in \Gamma(v)} y_u(i-1) < 0$ **then**
             $y_v(i) \leftarrow \epsilon$
         **else**
             $y_v(i) \leftarrow -1$
         **end**
     **else**
         $x_v(i) \leftarrow -\epsilon$ and $y_v(i) \leftarrow 0$
     **end**
**end**
`/* Computing the aggregating function ` $g(\{r(v) : v \in T\})$     `*/`
$\bar{y}_v \leftarrow \left( \sum_{i=1}^{\texttt{NumRounds}} (y_v(i) + 1)/(1 + \epsilon) \right) / \texttt{NumRounds}$
**return** $\sum_{v \in T} \bar{y}_v$

---

the cover, then $v$ is chosen to be in the cover; (3) if all of $v$'s neighbors are inactive, then $v$ remains inactive and no decision is made on $y_v$.

At the end of the local computation rounds, the final vertex cover size is computed by first averaging the $y_v$ time-series at each $v \in T$ (with translation, and scaling as shown in Algorithm 1), and then summing over all vertices.

---

**Algorithm 2:** Testing procedure of GRAPH2SEQ on a graph instance.

---

**Input:** graph $G$, trained parameters, objective $f : G \to \mathbb{R}$ we seek to maximize, maximum
      sequence length $T_{\max}$
**Output:** solution set $S \subseteq V$
$S_{\mathrm{opt}} \leftarrow \{\}, v_{\mathrm{opt}} \leftarrow 0$                                         `// initialize`
**for** $T$ *from* 1 *to* $T_{\max}$ **do**
    | $S \leftarrow$ solution returned by GRAPH2SEQ$(T)$
    | **if** $f(S) > v_{\mathrm{opt}}$ **then**
    |     | $S_{\mathrm{opt}} \leftarrow S$
    |     | $v_{\mathrm{opt}} \leftarrow f(S)$
    | **end**
**end**
**return** $S_{\mathrm{opt}}$

---

## C    SECTION 4 DETAILS

### C.1    REINFORCEMENT LEARNING FORMULATION

Let $G(V, E)$ be an input graph instance for the optimization problems mentioned above. Note that the solution to each of these problems can be represented by a set $S \subseteq V$. In the case of the minimum vertex cover (MVC) and maximum independent set (MIS), the set denotes the desired optimal cover and independent set respectively; for max cut (MC) we let $(S, S^c)$ denote the optimal cut. For the following let $f : 2^V \to \mathbb{R}$ be the objective function of the problem (i.e., MVC, MC or MIS) that we want to maximize, and let $\mathcal{F} \subseteq 2^V$ be the set of feasible solutions.

**Dynamic programming formulation.** Now, consider a dynamic programming heuristic in which the subproblems are defined by the pair $(G, S)$, where $G$ is the graph and $S \subseteq V$ is a subset of vertices that have already been included in the solution. For a vertex $a \in V \backslash S$ let $Q(G, S, a) = \max_{S' \supseteq S \cup \{a\}, S' \in \mathcal{F}} f(S') - f(S \cup \{a\})$ denote the marginal utility gained by selecting vertex $a$. Such a $Q$*-function* satisfies the Bellman equations given by

$$Q(G, S, a) = f(S \cup \{a\}) - f(S) + \max_{a' \in V \backslash S \cup \{a\}} Q(G, S \cup \{a\}, a'). \tag{22}$$

It is easily seen that computing the $Q$-functions solves the optimization problem, as $\max_{S \in \mathcal{F}} f(S) = \max_{a \in V} Q(G, \{\}, a)$. However exactly computing $Q$-functions may be computationally expensive. One approach towards approximately computing $Q(G, S, a)$ is to fit it to a (polynomial time computable) parametrized function, in a way that an appropriately defined error metric is minimized. This approach is called $Q$*-learning* in the reinforcement learning (RL) paradigm, and is described below.

**State, action and reward.** We consider a reinforcement learning policy in which the solution set $S \subseteq V$ is generated one vertex at a time. The algorithm proceeds in rounds, where at round $t$ the RL agent is presented with the graph $G$ and the set of vertices $S_t$ chosen so far. Based on this *state* information, the RL agent outputs an *action* $A_t \in V \backslash S_t$. The set of selected vertices is updated as $S_{t+1} = S_t \cup \{A_t\}$. Initially $S_0 = \{\}$. Every time the RL agent performs an action $A_t$ it also incurs a *reward* $R_t = f(S_t \cup \{A_t\}) - f(S_t)$. Note that the $Q$-function $Q(G, S_t, a)$ is well-defined only if $S_t$ and $a$ are such that there exists an $S' \supseteq S_t \cup \{a\}$ and $S' \in \mathcal{F}$. To enforce this let $\mathcal{F}_t = \{a \in V \backslash S_t : \exists S' \text{ s.t. } S' \supseteq S_t \cup \{a\} \text{ and } S' \in \mathcal{F}\}$ denote the set of feasible actions at time $t$. Each round, the learning agent chooses an action $A_t \in \mathcal{F}_t$. The algorithm terminates when $\mathcal{F}_t = \{\}$.

**Policy.** The goal of the RL agent is to learn a *policy* for selecting actions at each time, such that the cumulative reward incurred $\sum_{t \geq 0} R_t$ is maximized. A measure of the generalization capability of the policy is how well it is able to maximize cumulative reward for different graph instances from a collection (or from a distribution) of interest.

$Q$**-learning.** Let $\tilde{Q}(G, S, a; \Theta)$ denote the approximation of $Q(G, S, a)$ obtained using a parametrized function with parameters $\Theta$. Further let $((G_i, S_i, a_i))_{i=1}^N$ denote a sequence of (state,

action) tuples available as training examples. We define empirical loss as

$$\hat{L} = \sum_{i=1}^{N} \left( \tilde{Q}(G_i, S_i, a_i; \Theta) - f(S_i \cup \{a_i\}) + f(S_i) - \max_{a' \in V \setminus S_i \cup \{a_i\}} \tilde{Q}(G_i, S_i \cup \{a_i\}, a'; \Theta) \right)^2,$$

(23)

and minimize using stochastic gradient descent. The solution of the Bellman equations (22) is a stationary point for this optimization.

**Remark.** Heuristics such as ours, which select vertices one at a time in an irreversible fashion are studied as 'priority greedy' algorithms in computer science literature (Borodin et al., 2003; Angelopoulos & Borodin, 2003). The fundamental limits (worst-case) of priority greedy algorithms for minimum vertex cover and maximum independent set has been discussed in Borodin et al. (2010).

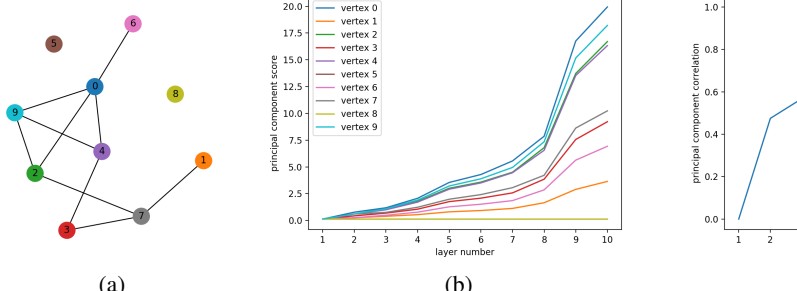

(a)                          (b)                          (c)

Figure 5: (a) Erdos-Renyi graph of size 10 considered in Figures (b) and (c), (b) vertex-wise principal component scores at each layer, and (c) projection of the principal direction at each iteration on the principal direction of iteration 10. These experiments are performed on our trained model.

## D    GEOMETRY OF ENCODING AND SEMANTICS OF GRAPH2SEQ

Towards an understanding of what aspect of solving the MVC is learnt by GRAPH2SEQ, we conduct empirical studies on the dynamics of the state vectors as well as present techniques and semantic interpretations of GRAPH2SEQ. This is done in the same spirit as in Section 3.3.

In the first set of experiments, we investigate the vertex state vector sequence. We consider graphs of size up to 50 and of types discussed in Section 5.2. For each fixed graph, we observe the vertex state $\mathbf{x}(\cdot)$ (Equation 2) evolution to a depth of 10 layers.

**(1) Dimension collapse.** As in the random parameter case, we observe that on an average more than 8 of the 16 dimensions of the vertex state become zeroed out after 4 or 5 layers.

**(2) Principal components' alignment.** The principal component direction of the vertex state vectors at each layer converges. Fig. 5(c) shows this effect for the graph shown in Fig. 5(a). We plot the absolute value of the inner product between the principal component direction at each layer and the principal component direction at layer 10.

**(3) Principal component scores and local connectivity.** The component of the vertex state vectors along the principal direction roughly correlate to how well the vertex is connected to the rest of the graph. We demonstrate this again for the graph shown in Fig. 5(a), in Fig 5(b). One significant difference here, from the random parameters case, is that the value of the principal score in the trained model do not rise as sharply as the random case (Fig. 1(b)) with increasing number of layers.

**(4) Optimal depth**. We study the effect of depth on approximation quality on the four graph types being tested (with size 50); we plot the vertex cover quality as returned by GRAPH2SEQas we vary the number of layers up to 25. Fig. 6(a) plots the results of this experiment, where there is no convergence behavior but nevertheless apparent that different graphs work optimally at different layer values. While the optimal layer value is 4 or 5 for random bipartite and random regular graphs, the worst case greedy example requires 15 rounds. This experiment underscores the importance of having a *flexible* number of layers is better than a fixed number; this is only enabled by the time-series nature of GRAPH2SEQ and is inherently missed by the fixed-depth GCNN representations in the literature.

**(5) $Q$-function semantics.** Recall that the $Q$-function of Equation (7) comprises of two terms. The first term, denoted by $Q_1$, is the same for all the vertices and includes a sum of all the $\mathbf{y}(\cdot)$ vectors. The second term, denoted by $Q_2(v)$ depends on the $\mathbf{y}(\cdot)$ vector for the vertex being considered. In this experiment we plot these two values at the very first layer of the learning algorithm (on a planted vertex cover graph of size 15, same type as in the training set) and make the following observations: (a) the values of $Q_1$ and $Q_2(\cdot)$ are close to being integers. $Q_1$ has a value that is one less than the negative of the minimum vertex cover size. (b) For a vertex $v$, $Q_2(v)$ is binary valued from the set $\{0, 1\}$. $Q_2(v)$ is one, if vertex $v$ is part of an optimum vertex cover, and zero otherwise. Thus the

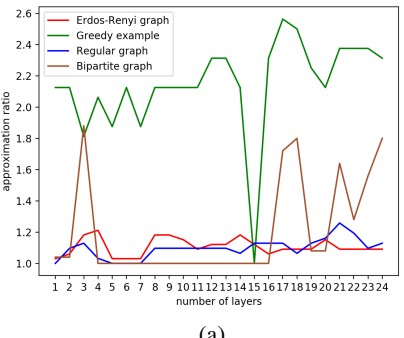 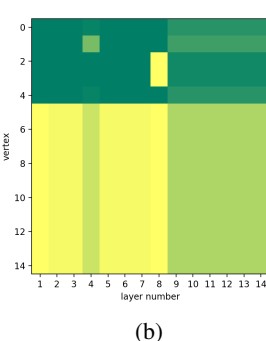 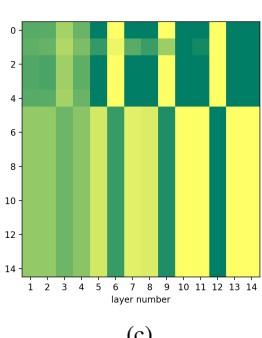

(a)                                    (b)                                    (c)

Figure 6: (a) Approximation ratio of GRAPH2SEQ with varying number of layers, (b) $\mathbf{y}(\cdot)$ vectors of GRAPH2SEQ in the intermediate layers seen using the $Q$-function, (c) $\mathbf{x}(\cdot)$ vectors of the fixed-depth model seen using the $Q$-function. Figure (b) and (c) are on planted vertex cover graph with optimum cover of vertices $\{0, 1, 2, 3, 4\}$.

neural network, in principle, computes the complete set of vertices in the optimum cover at the very first round itself.

**(6) Visualizing the learning dynamics**. The above observations suggests to 'visualize' how our learning algorithm proceeds in each layer of the evolution using the lens of the value of $Q_2(\cdot)$. In this experiment, we consider size-15 planted vertex cover graphs on (i) GRAPH2SEQ, and (ii) the fixed-depth GCNN trained on planted vertex cover graphs. Fig. 6(b) and 6(c) show the results of this experiment. The planted vertex cover graph considered for these figures has an optimal vertex cover comprising vertices $\{0, 1, 2, 3, 4\}$. We center (subtract mean) the $Q_2(\cdot)$ values at each layer, and threshold them to create the visualization. A dark green color signifies the vertex has a high $Q_2(\cdot)$ value, while the yellow means a low $Q_2(\cdot)$ value. We can see that in GRAPH2SEQ the heuristic is able to compute the optimal cover, and moreover this answer does not change with more rounds. The fixed depth GCNN has a non-convergent answer which oscillates between a complementary set of vertices. Take away message: having an upper LSTM layer in the learning network is critical to identify when an optimal solution is reached in the evolution, and "latch on" to it.

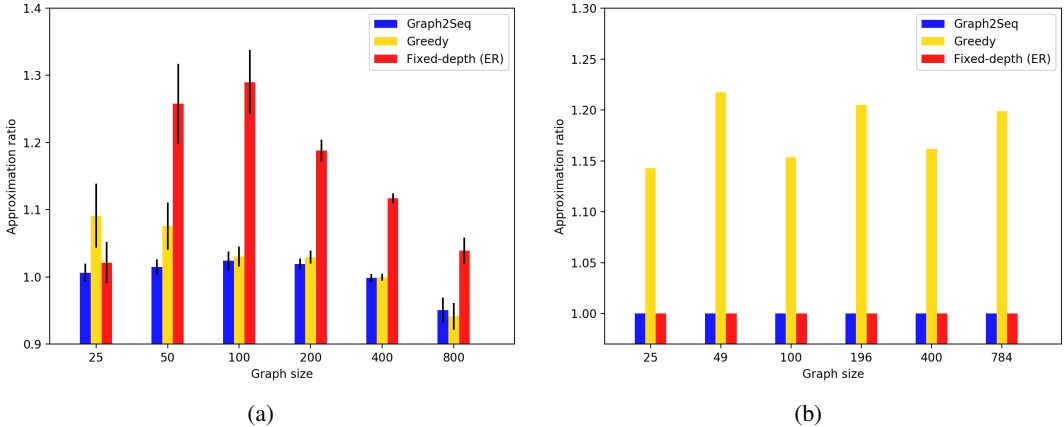

Figure 7: Max cut in (a) Erdos-Renyi graphs, (b) Grid graphs.

# E   EVALUATION: MAX CUT AND MAXIMUM INDEPENDENT SET

In this section we test and compare GRAPH2SEQ on the maximum cut and the maximum independent set problems. As in the MVC case, our results demonstrate a consistently good performance of GRAPH2SEQ across different graph types and sizes.

## E.1   MAX CUT

In the maximum cut problem, for an input graph instance $G(V, E)$ we seek a cut $(S, S^c)$ where $S \subseteq V$ such that the number of edges crossing the cut is maximized. This problem can be approximated within a factor 1.1383 of optimal, but not within 1.0684 unless $P = NP$. The RL model chooses vertices one at a time, with $c_v$ set to 1 whenever vertex $v$ is chosen. Supposing $S_t$ is the set of vertices chosen at the beginning of the $t$-th iteration. Then the reward $R_t$ for choosing a vertex $a \in V \backslash S_t$ is given by $R_t = |\{u : (u, a) \in E, u \in V \backslash S_t\}| - |\{u : (u, a) \in E, u \in S_t\}|$, i.e., the number of edges added to the cut because of moving $a$ from $V \backslash S_t$ less the number of edges lost. The vertex chosen $A_t$ is restricted to be in the set of feasible vertices $\mathcal{F}_t \subseteq V \backslash S_t$ for which the potential reward is non-negative. The algorithm terminates when $\mathcal{F}_t = \{\}$.

**Training.** We train our neural network model on size 15 random Erdos-Renyi graphs with an edge probability of 0.15. The technique for training and hyper-parameters are chosen to be the same as in Section 5.1.

The testing procedure is described below.

**Graph sizes and types.** We test on the following graphs.
(1) Erdos-Renyi graph, with edge probability $p = 0.15$.
(2) Two-dimensional grid graph, in which the sides contain equal number of vertices.
For each graph type, we vary the number of vertices in the range 25 – 800, and use the same trained model in all of the tests. The number of layers in GRAPH2SEQ is limited to 15.

**Heuristics compared.** We compare GRAPH2SEQ with the following two heuristics.
(1) Greedy, a simple algorithm that is a 2-approximation to the optimal. This algorithm proceeds in rounds, where in each round the value of the cut is strictly increased. Supposing $(S_t, V \backslash S_t)$ is the cut at round $t$ (initially $S_t = \{\}$). We choose an arbitrary vertex in either $S_t$ or $V \backslash S_t$ that, if switched to the other set strictly increases the value of the cut, and switch it. Note that in this algorithm a vertex could be selected and move between the sets more than once. In contrast our learning algorithm selects vertices at most once.
(2) Fixed-depth GCNN. We consider a depth-5 network, similar to the one considered in MVC, and train it on size-15 Erdos Renyi graphs with $p = 0.15$.
We report our results relative to an integer program solved using Gurobi, with a time cut-off of

240s. Since this is a maximization problem, we divide the Gurobi baseline solution to the heuristics compared in order to keep the approximation ratio greater than 1.

**Results.** The results of our tests are presented in Fig. 7. We notice that for both graph types GRAPH2SEQ is able to achieve an approximation less that $1.04$ times the (timed) integer program output. In Erdos-Renyi graphs of size greater than 400, it is even better than the integer program solution (due to the time cut-off).

The Greedy heuristic performs well on large size ($\geq 400$) Erdos-Renyi graphs, where it even beats GRAPH2SEQ by a small margin. However at small sizes it is at least 5% worse than GRAPH2SEQ in terms of approximation ratio. On the other hand, in grid graphs Greedy is consistently worse having an approximation ratio at least 15% above baseline values.

Fixed-depth GCNN also demonstrates a graph structure dependent behavior. On grid graphs it is similar to GRAPH2SEQ and achieves a consistent performance close to optimal. However on Erdos-Renyi graphs, starting with a good approximation ratio at size 25, it quickly worsens to more than 25% above optimal at size 100. This is surprising considering that the heuristic is also trained on Erdos-Renyi graphs.

### E.2 INDEPENDENT SET

For an unweighted graph $G(V, E)$ the maximum independent set denotes a set $S \subseteq V$ of maximum cardinality such that for any $u, v \in S$, $(u, v) \notin E$. The maximum independent set is complementary set of vertices to the minimum vertex cover, i.e., if $S$ is a maximum independent set of $G$, then $V \backslash S$ is the minimum vertex cover. However, from an approximations point-of-view maximum independent set is hard to approximate within $n^{1-\epsilon}$ for any $\epsilon > 0$, despite constant factor approximation algorithms known for minimum vertex cover.

We follow a similar reinforcement learning model, as in the other evaluations. Let $S_t \subseteq V$ denote the set of vertices selected before the $t$-th round (initially $S_1 = \{\}$). Let $\mathcal{F}_t = \{u : u \in V \backslash S_t, (u, v) \notin E \ \forall v \in S_t\}$ denote the set of vertices that can be added to $S_t$ without violating the independent set property of $S_t$. The learning model chooses a vertex $a \in \mathcal{F}_t$ having the largest $Q$-value, as action at round $t$. This incurs a reward of $R_t = +1$ and $S_t$ is updated as $S_{t+1} = S_t \cup \{a\}$. The algorithm terminates when $\mathcal{F}_t = \{\}$.

**Training.** As in max cut, we train on size-15 Erdos-Renyi graphs with edge probability of 0.15. The other hyper-parameters are set to be the same as in the minimum vertex cover evaluation (Section 5.1).

**Heuristics compared.** We compare GRAPH2SEQ with the following heuristics.
(1) Greedy. This is a simple heuristic which achieves a $\Delta + 1$ approximation on unweighted graphs with maximum degree of $\Delta$. As with the other greedy heuristics, the heuristic proceeds in rounds selecting one vertex in each round. Let $S_t \subseteq V$ denote the vertices chosen before round $t$ ($S_1 = \{\}$). For input graph $G$, we first construct a graph $\tilde{G}_t$ which is the same as $G$ but with $S_t$ and its neighboring vertices removed. The Greedy heuristic chooses as its selection a vertex having the least degree in $\tilde{G}_t$ in round $t$. The heuristic terminates when $\tilde{G}_t$ is empty.
(2) Fixed-depth GCNN, with 5 layers and trained on size-15 Erdos-Renyi graphs with an edge probability $p = 0.15$. The training is performed the same way as in minimum vertex cover and max cut evaluations.
As baseline we use the Gurobi optimization package to compute the optimal integer programming solution with a time cut-off of 240s. We divide the Gurobi baseline solution to the heuristics compared in order to keep the approximation ratio greater than 1 as in max cut.

**Graph types and sizes.** The following graphs are considered.
(1) Erdos-Renyi graphs generated with edge probability $p = 0.15$.
(2) Structured bipartite graphs. We construct an adversarial class of bipartite graphs for the Greedy heuristic. For a parameter $m \in 2\mathbb{N}$ we will describe a construction having $3m/2 + 2$ vertices. The maximum independent set size is $m$, while Greedy chooses a set of size $m/2 + 2$ giving an approximation ratio close to 2. Note that the construction below can be easily modified to generate arbitrarily high ($> 2$) constant approximation ratios for Greedy. The first partite consists of $m$ vertices $\{u_1, \ldots, u_m\}$ and the second partite consists of $m/2 + 2$ vertices $\{v_1, \ldots, v_{m/2+2}\}$. For

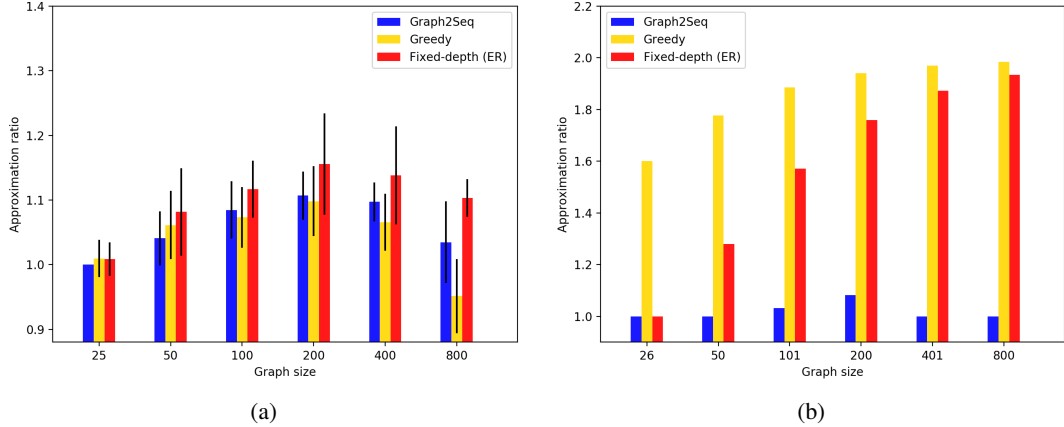

Figure 8: Maximum independent set in (a) Erdos-Renyi graphs, (b) structured bipartite graphs (described under graph types in Section E.2).

$i = 1, \ldots, m/2$ each $v_i$ is connected to $u_{2i-1}$ and $u_{2i}$. Whereas $v_{m/2+1}$ and $v_{m/2+2}$ are connected to all of $\{u_1, \ldots, u_m\}$.

For both graph types, we vary the total number of vertices in the range $25 - 800$. The number of layers in GRAPH2SEQ is restricted to 15 for Erdos-Renyi graphs, and 40 for the structured bipartite graph.

**Results.** We present our results in Fig. 8. In Erdos-Renyi graphs, GRAPH2SEQ shows a reasonable consistency in which it is always less than 1.10 times the (timed) integer program solution. The highest approximation ratio occurs with moderate graph sizes $(100 - 400)$. But even in this regime it is comparable to the other two heuristics. In the bipartite graph case we see a performance within 8% of optimal across all sizes.

Greedy heuristic does very well on Erdos-Renyi graphs, with comparable performance to GRAPH2SEQ up to moderate sizes, and even beating GRAPH2SEQ at large $(> 400)$ sizes. However, on the bipartite graphs it does poorly as expected.

The fixed-depth heuristic also does moderately well on Erdos-Renyi graphs (though it is worse than both GRAPH2SEQ and Greedy at all sizes). However as with Greedy, it progressively becomes worse (close to an approximation ratio of 2) on the bipartite graphs.

