# OpenReview forum: "Graph2Seq: Scalable Learning Dynamics for Graphs"
_ICLR.cc/2018/Conference — Reject_

### Official Review · AnonReviewer3 · 2017-11-27
**The motivation for this paper is unclear. The writing is problematic and evaluations are not sufficient.**

**Rating:** 4
**Confidence:** 4

**Review:**

This paper proposes to represent nodes in graphs by time series. This is an interesting idea but the results presented in the paper are very preliminary.
Experiments are only conducted on synthetic data with very small sizes.
In Section 5.1, I did not understand the construction of the graph. What means 'all the vertices are disjoint'? Then I do not understand why the vertices of G_i form the optimum.

---

> ### Author Response · Authors · 2018-01-02
> **Graph construction**
>
> Vertices are disjoint means the vertices do not have any edge between them. Since the vertices of G_o do not have any edge between themselves, selecting the vertices of G_i as a cover will ensure every edge of the graph is covered.

---

### Official Review · AnonReviewer2 · 2017-12-01
**Interesting idea but limited validation. Also sample complexity may be exponential in graph degree.**

**Rating:** 4
**Confidence:** 4

**Review:**

This paper proposes a novel way of embedding graph structure into a sequence that can have an unbounded length.

There has been a significant amount of prior work (e.g. d graph convolutional neural networks) for signals supported on a specific graph. This paper on the contrary tries to encode the topology of a graph using a dynamical system created by the graph and randomization.

The main theorem is that the created dynamical system can be used to reverse engineer the graph topology for any digraph.
As far as I understood, the authors are doing essentially reverse directed graphical model learning. In classical learning of directed graphical models (or causal DAGs) one wants to learn the structure of a graph from observed data created by this graph inducing conditional independencies on data. This procedure is creating a dynamical system that (following very closely previous work) estimates conditional directed information for every pair of vertices u,v and can find if an edge is present from the observed trajectory.
The recovery algorithm is essentially previous work (but the application to graph recovery is new).

The authors state:
``Estimating conditional directed information efficiently from samples is itself an active area of research Quinn et al. (2011), but simple plug-in estimators with a standard kernel density estimator will be consistent.''

One thing that is missing here is that the number of samples needed could be exponential in the degrees of the graph. Therefore, it is not clear at all that high-dimensional densities or directed information can be estimated from a number of samples that is polynomial in the dimension (e.g. graph degree).

This is related to the second limitation, that there is no sample complexity bounds presented only an asymptotic statement.

One remark is that there are many ways to represent a finite graph with a sequence that can be decoded back to the graph (and of course if there is no bound on the graph size, there will be no bound on the size of the sequence). For example, one could take the adjacency matrix and sequentially write down one row after the other (perhaps using a special symbol to indicate 'next row'). Many other simple methods can be obtained also, with a size of sequence being polynomial (in fact linear) in the size of the graph. I understand that such trivial representations might not work well with RNNs but they would satisfy stronger versions of Theorem 1 with optimal size.
On the contrary it was not clear how the proposed sequence will scale in the graph size.


Another remark is that it seems that GCNN and this paper solve different problems.
GCNNs want to represent graph-supported signals (on a fixed graph) while this paper tries to represent the topology of a graph, which seems different.


The experimental evaluation was somewhat limited and that is the biggest problem from a practical standpoint. It is not clear why one would want to use these sequences for solving MVC. There are several graph classification tasks that try to use the graph structure (as well as possibly other features) see eg the bioinformatics
and other applications. Literature includes for example:
Graph Kernels by S.V.N. Vishwanathan et al.
Deep graph kernels (Yanardag & Vishwanathan and graph invariant kernels (Orsini et al.),
which use counts of small substructures as features.

The are many benchmarks of graph classification tasks where the proposed representation could be useful but significantly more validation work would be needed to make that case.

---

> ### Author Response · Authors · 2018-01-02
> **Sequence lengths**
>
> We first note that recovering the graph topology from the time-series is not the primary objective of Graph2Seq (we already have the graph as our input, there is no need to recover it). The main goal of Graph2Seq is to provide a representation framework for learning tasks (e.g., classification, optimization), over graphs that are not fixed.
>
> Supposing we have a candidate neural network framework (such as Graph2Seq) that can take in arbitrary sized graphs as input, and produce an output. Knowing whether such a framework could work well on graphs of any size is unfortunately a difficult question to answer. In this context, we have included Theorem 1 as a strong conceptual evidence towards the scalability of Graph2Seq. The fact that the entire graph topology can be recovered from the Graph2Seq representation (even if we ignore sample complexity and computation issues) suggests the time-series has enough information to recover the graph in principle.
>
> Indeed, there are many ways in which one could represent a graph as a sequence (with potentially shorter sequences). However, the issue with methods involving the adjacency matrix is they require a prior labelling of the graph nodes (to identify the individual rows and columns of the matrix), and it is not clear how to incorporate such labels into the neural network. This is perhaps why the adjacency matrix is itself not used as a representation in the first place, and methods like GCNN are necessary. What we are seeking is a label-free representation.

---

> > ### Comment · AnonReviewer2 · 2018-01-20
> > **response of reviewer**
> >
> > I read the response and I do not feel I should change my review since mostly my concerns remain.
> >
> > The authors did not acknowledge that their sequence representation can be exponential length, or if I am mistaken ?
> >
> > As a general remark, if you could map a graph into a poly-size sequence that is invariant to labeling of the graph nodes and this sequence is invertible (i.e you can use it to reconstruct the graph) then you have solved graph isomorphism.
> > This is because two graphs would be isomorphic iff their sequences are identical.

---

> > > ### Author Response · Authors · 2018-01-26
> > > **our response**
> > >
> > > (1) Reg. length of sequence:
> > >
> > > Sequence length depends on the graph, and in the worst case is exponential in # of edges. In any case, this is irrelevant to our (empirical) results. In fact, the theoretical fact of exponential dependence in # of edges only makes our empirical results more impressive; we only need to use sequence lengths roughly equal to the diameter of the graph to get our numerical results, which are favorably competitive with the best algorithms (not just prior neural network methods) for the problems studied.
> > >
> > > (2) Reg. the general remark:
> > >
> > > The remark holds true only under deterministic sequence generation.
> > >
> > > Under deterministic initialization and evolution, the sequence cannot be used to distinguish even non-isomorphic graphs, as we have showed in the proof of Proposition 1 in the paper. This is a clear limitation of deterministic sequence generation. We point this out in Section 3.1.
> > >
> > > However, if the evolution is random (by adding a random node label or noise), then the sequences are no longer identical even for isomorphic graphs, and as such cannot be used as a test for isomorphism.

---

### Official Review · AnonReviewer1 · 2017-12-05
**Unclear motivation and experiments**

**Rating:** 4
**Confidence:** 3

**Review:**

The paper proposes GRAPH2SEQ that represents graphs as infinite time-series of vectors, one for
each vertex of the graph and in an invertible representation of a graph.  By not having the restriction of representation to a fixed dimension, the authors claims their proposed method is much more scalable. They also define a formal computational model, called LOCAL-Gather that includes GRAPH2SEQ and other classes of GCNN representations, and show that GRAPH2SEQ is capable of computing certain graph functions that fixed-depth GCNNs cannot. They experiment on graphs of size at most 800 nodes to discover minimum vertex cover and show that their method perform much better than GCNNs but is comparable with greedy heuristics for minimum vertex cover.

I find the experiments to be hugely disappointing. Claiming that this particular representation helps in scalability and then doing experiment on graphs of extremely small size does not reflect well. It would have been much more desirable if the authors had conducted experiments on large graphs and compare the results with greedy heuristics. Also, the authors need to consider other functions, not only minimum vertex cover. In general, lack of substantial experiments makes it difficult to appreciate the novelty of the work. I am not at all sure, if this representation is indeed useful for graph optimization problems practically.

---

> ### Author Response · Authors · 2018-01-02
> **On graph size**
>
> We have conducted experiments on graphs of size up to 3200, and will include in our revision. Graph2Seq’s performance trend continues to hold at this size. We also tried larger graph sizes, but due to the large number of edges we ran into computational and memory issues (25k and 100k size graphs, which have 46 million and 4 billion edges respectively). Even doing greedy algorithms at this scale is computationally hard. As mentioned previously, our test graphs are not sparse and the current test graphs contain a large number of edges (hundreds of thousands to a million). We also reiterate that our training is on graphs of size 15, illustrating a generalization over a factor of 200. Evaluations for maximum independent set and max cut functions have been included in the appendix.

---

### Author Response · Authors · 2018-01-02
**Response to reviewers**

We thank the reviewers for the helpful comments. Please find our response to the issues raised below.

On motivation:

We are rather puzzled by the comment that the motivations are unclear. Using neural networks for graph structured data is a fast-emerging field and is of topical interest (massive attendance in a recent NIPS workshop on Non-Euclidean deep learning https://nips.cc/Conferences/2017/Schedule?showEvent=8735 serves to illustrate). Our paper directly addresses one of the key open problems in the area: how to design neural networks for graphs that can scale to graph inputs of arbitrary sizes and shapes.

Such a scalable solution may be required for a variety of reasons: (1) directly training on large instances may not be possible; (2) application specific training can be avoided, and trained algorithms can be used in variety of settings; or (3) a scalable algorithm may be easier to analyze, reason about and can potentially inspire advances in theory CS. Indeed, traditionally algorithms in CS have usually been of this flavor. However, to our best awareness, such an analog in deep learning for graphs has been critically missing.

The combinatorial optimization problems we have used in our evaluations (vertex cover, max cut, max independent set) are also interesting and many recent works (e.g. Bello et al ’17, Vinyals et al ’15, Dai et al ‘17) have considered these problems. Moreover, input instances in these problems capture the very essence of what makes representing signals over non-fixed graphs challenging: (i) the input graphs could have arbitrary topology, and (ii) the input graphs could have arbitrary size. The simplicity of these problems (in terms of vertex/edge features) allow us to focus on directly addressing these two scalability issues without worrying about dependencies arising from high-dimensional node/edge attributes.

On evaluations:

We have evaluated graphs of size up to 3200 and will include in our revision. Our test graphs are not sparse, and contain a large number of edges: e.g., a 3200 node Erdos-Renyi graph has 700,000 edges; a 3200 node random bipartite graph has 1.9 million edges. These graph sizes are consistent and well-above the sizes used in the neural networks combinatorial optimization literature (e.g., Learning combinatorial optimization algorithms over graphs, Dai et al, NIPS ’17 (up to 1200 nodes); Neural combinatorial optimization with reinforcement learning, Bello et al, ’17 (100 nodes); Pointer networks, Vinyals et al, NIPS ’15 (50 nodes)). Compared to the recent NIPS spotlight paper by Dai et al (which focuses on similar combinatorial problems), our results illustrate significant generalizations both in graph topology, and graph size.

The space of problems where the graph instances are not fixed is vast, and finding scalable learning representations for these applications remains a grand challenge. To our knowledge, this is also a longer-term project and a one-size-fits-all approach that solves all of those applications may not be possible. In this regard, our work presents an important first-step of recognizing, formalizing and understanding the key challenges involved, and also proposes a promising solution that directly addresses the key issues.

---

### Decision · Program_Chairs · 2018-01-29
**ICLR 2018 Conference Acceptance Decision**

**Decision:**

Reject

**Comment:**

The reviewers agree that the problem being studied is important and relevant but express serious concerns. I recommend the authors to carefully go through the reviews and significantly scale up their experiments.